# Dissipative realization of Kondo models

Martino Stefanini [1], Yi-Fan Qu [2] ✉, Tilman Esslinger [3], Sarang Gopalakrishnan[4],
Eugene Demler [2] & Jamir Marino[1]

The Kondo effect is a prototypical strongly correlated phenomenon, and it is usually discussed in the context of unitary dynamics. Here, we demonstrate that the Kondo effect can be induced through non-linear dissipative channels, without requiring any coherent interaction on the impurity site. Specifically, we consider a reservoir of noninteracting fermions that can hop on a few impurity sites that are subjected to strong two-body losses. In the simplest case of a single lossy site, we recover the Anderson impurity model in the regime of infinite repulsion, with a small residual dissipation as a perturbation. While the Anderson model gives rise to the Kondo effect, this residual dissipation competes with it, offering an instance of a nonlinear dissipative impurity where the interplay between coherent and incoherent dynamics emerges from the same underlying physical process. We further outline how this dissipative engineering scheme can be extended to two or more lossy sites, realizing generalizations of the Kondo model with spin 1 or higher. Our results suggest alternative implementations of Kondo models using ultracold atoms in transport experiments, where localized dissipation can be naturally introduced, and the Kondo effect observed through conductance measurements.

The Kondo effect[1] is one of the simplest and most iconic phenomena in the physics of strongly correlated systems. It emerges when an interacting impurity exchanges particles with a gapless fermionic reservoir. The hybridization of the impurity levels with the bath's states causes the emergence of a very narrow many-body resonance (the Kondo, or Abrikosov-Suhl resonance) pinned at the chemical potential of the reservoir, whose properties dominate the low-energy physics and lead to a number of fascinating phenomena, such as universal scaling behavior of thermodynamics quantities[1–12] in impure metals, and an almost perfect conductance through the impurity in quantum dots[13–19]. From a theoretical perspective, the Kondo effect is a source of enduring interest as it showcases how strongly correlated behavior can emerge from simple ingredients.

With the advent of quantum simulation with ultracold atomic gases, new possibilities have arisen for the study of strongly correlated physics in regimes that would be otherwise inaccessible to traditional solid-state set-ups, and consequently, there have been many proposals for realizing the Kondo effect[20–34]. Such realization would be desirable for accessing less understood properties of the Kondo model and its relatives, such as their nonequilibrium dynamics, including the spreading of correlations in real space—i.e., the formation of the Kondo screening cloud, which has eluded measurement in conventional platforms until recently[35]. Moreover, realizing the Kondo models would provide a stepping stone for implementing more complicated models that are harder to realize in solid state setups, such

as high-spin and multi-channel Kondo models[1,12,36–46], or whose theoretical understanding is more limited, such as the Kondo lattice that is used to model heavy fermion compounds[1,3,47,48]. Despite the numerous proposals, the Kondo effect is yet to be observed with ultracold atoms. Only the first steps in achieving a Kondo lattice configuration have been performed so far[49].

In this work, we show how a strong, localized two-body loss[50–54] within a noninteracting fermionic gas can provide the correlations necessary to induce the Kondo effect. The significance of our results is twofold. On the one hand, we provide a physically motivated example of a nonlinear dissipative impurity system, in which the many-body nature of dissipation imprints correlations on the system. Indeed, we observe a competition between the incoherent effect of losses and the coherent, Kondo dynamics that they induce. The present work complements the growing corpus of literature that has been devoted to linear impurities, i.e., featuring single-body losses[55–67] or gains[68], as well as local dephasing[69,70]. On the other hand, the present work points to a dissipative route to the implementation of both standard and more exotic Kondo models in ultracold atoms. An appealing feature of this route is that losses can be easily incorporated in the recently developed transport experiments with ultracold atoms[60–64,69,71–73] that mimic the configuration of mesoscopic systems like quantum dots. This setup introduces the possibility of revealing the emergence of the Kondo effect in the same, well-established way of quantum dots, namely via conductance

[1]Institut für Physik, Johannes Gutenberg-Universität Mainz, Mainz, Germany. [2]Institute for Theoretical Physics, ETH Zürich, Zurich, Switzerland. [3]Institute for Quantum Electronics & Quantum Center, ETH Zürich, Zurich, Switzerland. [4]Department of Electrical and Computer Engineering, Princeton University, Princeton, NJ, USA. ✉e-mail: yifaqu@phys.ethz.ch

measurements. This approach is rather direct and may prove to be more sensitive than direct measurements of the spectral function through radio-frequency spectroscopy[74].

We remark that our results regard the emergence of a typically Hamiltonian effect by means of dissipation. In this, we are distinguished from other recent works that use dissipation to introduce entirely new features, such as engineering of non-Hermitian versions of the Kondo model[75–77] and measurement-induced crossovers in continuously monitored quantum dots[76,78].

Our main finding is the characterization of the Kondo effect induced by a strong two-body loss localized on a single site connected to two reservoirs of noninteracting fermions. We argue that for an infinitely strong dissipation, this system realizes the well-known Anderson impurity model (AIM) with infinite repulsion, and we derive the leading corrections to the Lindblad master equation for finite dissipation rate, which take the form of a residual two-body loss. We analyze the typical signatures of the Kondo effect (Kondo resonance in the spectral function, enhanced differential conductance at zero bias and suppressed decay of magnetization) both in dynamics and in the local steady state, and we observe a competition between the Kondo effect and the residual losses, with the latter suppressing the former as the dissipation rate is decreased.

We conclude by describing how to realize higher-spin Kondo models by distributing the dissipation on more than one site, provided they are strongly coupled among themselves, and we briefly discuss the extension of the model to a multi-channel scenario.

## Results and discussion
### Model
We consider the system depicted in Fig. 1a, composed of two reservoirs (or leads) of noninteracting, spinful fermions that are connected to a central region (the "dot") whose sites are subjected to a two body loss—whenever two opposite-spin fermions occupy one of the dissipative sites, they can both be lost from the system with a certain rate $\gamma$. In principle, one could also include one- and three-body losses, but these processes are weak in a noninteracting Fermi gas, and can be neglected with respect to the large two-body losses assumed here. We are going to comment on the many-site and many-leads scenario at the end of the paper. In the rest of this work, we will focus on the simplest case of a single dot site hosting a single orbital, and coupled to two reservoirs. In the conditions common to experiments with ultracold atoms[50–52,60] losses are Markovian, and the dynamics of the system's density matrix $\rho(t)$ can be described by a Lindblad master equation ($\hbar = 1$)

$$\frac{\mathrm{d}}{\mathrm{d}t}\rho(t) = -\mathrm{i}\big[H, \rho(t)\big] + \gamma\left(L\rho(t)L^\dagger - \frac{1}{2}\{L^\dagger L, \rho(t)\}\right) \quad (1)$$

with jump operator $L = d_\downarrow d_\uparrow$ (where $d_\sigma$ annihilates a fermion with spin $\sigma$ on the dot, with $\sigma \in \{\uparrow, \downarrow\} = \{+, -\}$). The Hamiltonian has the familiar form of a resonant level model[79]

$$H = H_d + H_{\text{leads}} + H_{\text{tun}}$$
$$= \varepsilon_d \sum_\sigma d_\sigma^\dagger d_\sigma + \sum_{p\sigma\alpha} \varepsilon_{p\alpha} c_{p\sigma\alpha}^\dagger c_{p\sigma\alpha} + \sum_{p\sigma\alpha}(V_{p\alpha} d_\sigma^\dagger c_{p\sigma\alpha} + \text{H.c.}). \quad (2)$$

where we introduced the dot energy $\varepsilon_d$, $c_{p\sigma\alpha}$ annihilates a fermion in lead $\alpha \in \{R, L\}$ with momentum $p$ and single-particle energy $\varepsilon_{p\alpha} = \varepsilon_p - \mu_\alpha$, possibly biased by a chemical potential $\mu_\alpha$. We notice that the tuning of $\varepsilon_d$ and the possibility of biasing the reservoirs are both within the current experimental capabilities[71,80]. As it is usual in impurity problems, the leads can always be considered to be one-dimensional[1,37,38,79]. However, in a realistic transport setup, they are typically three-dimensional[71,81]. The tunneling between the dot and the leads is governed by the amplitudes $V_{p\alpha}$. For most purposes, the leads are fully described by the level width function[79] $\Gamma_\alpha(\omega) \equiv 2\pi\sum_p|V_{p\alpha}|^2\delta(\omega - \varepsilon_p) \equiv \Gamma_\alpha\xi(\omega)$, which we parametrize in terms of the energy scale $\Gamma_\alpha$, which contributes to the tunneling rate $\Gamma_T \equiv \sum_\alpha\Gamma_\alpha$ of the dot levels in the absence of dissipation, and the shape function $0 \leq \xi(\omega) \leq 1$ that sets the bandwidth $W \gg |\varepsilon_d|, \Gamma_\alpha$. We are going to assume symmetric leads $\Gamma_\alpha = \Gamma$ and a flat shape function $\xi(\omega) = \theta(W - |\omega|)$ ($\theta$ being the Heaviside function). We highlight that by assumption, no interactions are present in any portion of the system, so that the only source of correlations is the two-body losses. This simplifying assumption is not particularly restrictive, as interactions in the leads can be tuned via Feshbach resonances[71]. Moreover, we expect our model to be at least qualitatively valid even for weakly repulsive interactions, in the spirit of Fermi liquid theory (since realistic leads are three-dimensional).

We are interested in the regime of strong dissipation, in which the decay rate $\gamma$ is the largest energy scale. We will consider both the dynamics of the system after turning on the dissipation and the properties of the local steady state that forms around the dot at long times. Since the dissipation removes pairs of opposite-spin fermions, it is clear that in the limit $\gamma \to +\infty$ the system remains confined to a subspace with no double occupancies on the dot site(s). In the case of a single site, this subspace is spanned by three dark states, namely states which are simultaneously eigenstates of $H_{\text{dot}}$ and annihilated by the jump $L$: the empty dot $|0\rangle$ and the singly occupied dot $|\sigma\rangle$. The dynamics within the dark subspace is generated by the exchange of particles with the leads, according to the Hamiltonian:

$$H_{\text{eff}} = \varepsilon_d \sum_\sigma X_{\sigma\sigma} + \sum_{p\sigma\alpha} \varepsilon_{p\alpha} c_{p\sigma\alpha}^\dagger c_{p\sigma\alpha} + \sum_{p\sigma\alpha}(V_{p\alpha} X_{\sigma0} c_{p\sigma\alpha} + \text{H.c.}), \quad (3)$$

where the operators $X_{\sigma\sigma} \equiv |\sigma\rangle\langle\sigma|$, $X_{\sigma0} = X_{0\sigma}^\dagger \equiv |\sigma\rangle\langle0|$ and $X_{00} \equiv |0\rangle\langle0|$ are known as Hubbard operators[3]. The Hamiltonian (3) is the well-known

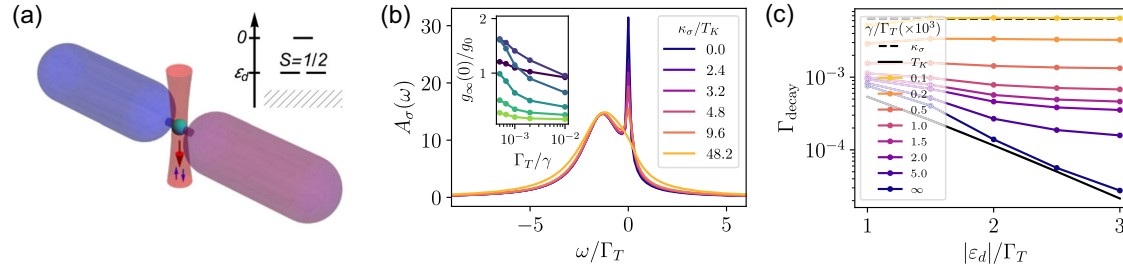

**Fig. 1 | Using strong, localized two-body losses to realize the Anderson impurity model with infinite repulsion. a** sketch of the simplest dissipative setup considered in the main text. The two tubes represent reservoirs of noninteracting, spinful fermions (possibly at different chemical potentials) that are allowed to tunnel to the central site, which is subjected to strong two-body losses. Inset: sketch of the dark states of the isolated impurity subjected to the two-body loss. The hatched area represents the dissipated states. **b** impurity spectral function in the local steady state, showing the smearing of the Kondo resonance as the effective dissipation is

increased. The parameters are $\varepsilon_d = -2\Gamma_T$ and $\Gamma_T = 10^{-2}W$ (we take the half-bandwidth $W$ as our energy unit), for unbiased reservoirs at $\mu = 0$ and zero initial temperature. Inset: zero-bias conductance across the impurity, normalized to $g_0 = 1/h$ ($h$ being Planck's constant). Lighter colors correspond to larger $|\varepsilon_d|/\Gamma_T \in \{0.5, 1, 1.5, 2, 2.5, 3\}$. **c** decay rate of the impurity magnetization, showing a crossover from the Kondo regime $\Gamma_{\text{decay}} \sim T_K$ to the incoherent regime $\Gamma_{\text{decay}} \sim \kappa_\sigma$. In these plots $\Gamma_T = 10^{-2}W$. In **b** and **c** we use the definition of $T_K$ reported in the main text, $T_K \equiv (\Gamma_T W/2)^{1/2} \exp[-\pi(\mu - \varepsilon_d)/\Gamma_T]$.

Anderson impurity model (AIM) with infinite repulsion[1,3,10,36], a strongly correlated model whose properties have been extensively studied. In particular, the model is known to show the Kondo effect—namely, its low-energy physics is dominated by a narrow many-body resonance pinned at the chemical potential of the leads.

It is important to understand to what extent the Kondo phenomenology associated with the AIM survives once we move away from the ideal $\gamma \to +\infty$ limit. As long as $\gamma$ is still the largest energy scale of the system, we can derive corrections to the dynamics in the constrained subspace by means of an adiabatic elimination[82] or, equivalently, via a dissipative generalization of the Schrieffer-Wolff transformation[83], the details of which are reported in Supplementary Note 1. Physically, in the presence of a strong dissipation $\gamma \gg \Gamma$, any double occupancy on the dot site is rapidly removed from the system before it can be replenished from the leads during a transient that lasts a few $\gamma^{-1}$. The dynamics at later times are effectively projected on the subspace of no double occupancies, in which it is described by the master equation

$$\frac{\mathrm{d}}{\mathrm{d}t}\rho(t) = -\mathrm{i}[H_{\mathrm{eff}}, \rho(t)] + L_{\mathrm{eff}}\rho(t)L_{\mathrm{eff}}^{\dagger} - \frac{1}{2}\{L_{\mathrm{eff}}^{\dagger}L_{\mathrm{eff}}, \rho(t)\}, \quad (4)$$

where the Hamiltonian (3) governs the coherent part, while there is a residual dissipation with the jump operator

$$L_{\mathrm{eff}} \equiv \frac{2}{\gamma^{1/2}}\sum_{p\sigma\alpha}\sigma V_{p\alpha}X_{0\bar{\sigma}}c_{p\sigma\alpha}. \quad (5)$$

The effective dissipation accounts for the virtual processes in which a fermion with spin $\sigma$ hops from a lead to the dot while the latter is already occupied by a fermion of opposite spin $\bar{\sigma}$, causing both fermions to be lost from the system due to the dissipation.

The effective master equation (4) introduces two new energy scales into the problem: the Kondo temperature $T_K \equiv (\Gamma_T W/2)^{1/2}$ $\exp[-\pi(\mu - \varepsilon_d)/\Gamma_T]$[1,3,84], governing the long-time behavior of the unitary dynamics generated by $H_{\mathrm{eff}}$ (with some exceptions, depending on the initial state[85]; the regimes of model (4) analyzed here do not seem to belong to the exceptional cases), and the residual loss rate, that at the leading order in $\Gamma_\alpha/\gamma$ and for a flat width function $\xi(\omega) = \theta(W - |\omega|)$ is given by

$$\kappa_\sigma = 2\sum_\alpha \frac{\Gamma_\alpha}{\pi\gamma}(\mu_\alpha + W). \quad (6)$$

For a derivation of this rate, see the Supplementary Note 2. In general, $\kappa_\sigma$ is non-universal in the sense that it depends on the specific band shape $\xi(\omega)$, but can be estimated as $\kappa_\sigma \sim \mathcal{O}(\Gamma_T W/\gamma)$—roughly speaking, it is proportional to the total density of fermions in the reservoirs. Since the Kondo effect is generated by coherent tunneling processes, we expect the residual dissipation to compete against it, i.e., that a lowering of $\gamma$ (which increases $\kappa_\sigma$) should wash away the Kondo effect. Indeed, in the limit of a vanishing $\gamma$ we recover the noninteracting model (2), which features no Kondo effect.

The effective master equation (4) has been derived assuming that $\gamma \gg W, |\varepsilon_d|, \Gamma$. We can further estimate its regime of validity by computing the probability of a double occupancy on the dot. As shown in the Supplementary Note 2, we obtain $\delta \equiv \langle d_\uparrow^\dagger d_\uparrow d_\downarrow^\dagger d_\downarrow \rangle \sim \mathcal{O}(\kappa_\sigma/\gamma) \sim \mathcal{O}(\Gamma_T W/\gamma^2)$. If we impose a somewhat arbitrary threshold $\delta \lesssim 10^{-2}$ for neglecting double occupancies, we obtain that $\gamma \gtrsim 10(\Gamma_T W)^{1/2}$. Since in our calculations we take $\Gamma_T \sim 10^{-2}W$, we can see that $\gamma$ can be almost as low as $W$.

## Signatures of Kondo

For a strictly infinite dissipation, the dynamics are governed by the effective AIM Hamiltonian (3), which guarantees the presence of Kondo physics—such as the Kondo resonance in the impurity spectral function and the maximal differential conductance at zero bias. Therefore, a first important task is to assess to which a finite but large value of $\gamma$ alters the well-known

Kondo features. A second interesting question is how these features emerge from the uncorrelated system at $\gamma = 0$ once the dissipation is increased.

During the years, a large number of different numerical techniques have been employed to tackle the dynamics of the AIM and of the Kondo model for isolated systems—to mention only some of the most recent, we can list methods based on the time-dependent variational principle[86,87], on the influence functional[88] and on matrix product states[89]. In the present work, we need to deal with dissipative dynamics, and we employ two different numerical approaches. One is based on quantum trajectories[90] and on a variational Ansatz for the state of the system along the trajectories[91,92]. The detailed description of this method and its main results is the subject of the companion paper[93]. The advantage of this method is that it can be applied for any value of $\gamma$, so that we can observe the full crossover towards the Kondo phenomenology as the dissipation is increased. The main observables accessible to the method are currents and single-time averages, and correlation functions.

In the present work, we take an alternative approach and work directly with the effective model (3) and the residual dissipation $L_{\mathrm{eff}}$ to observe the effects of a large but finite $\gamma$. In this regime, we use the slave boson representation[3,94,95] and we apply the widely used non-crossing approximation (NCA)[1,15,36,84,96–98] to derive Kadanoff-Baym equations for the Keldysh Green's functions of the auxiliary particles. To the same leading perturbative order as the NCA self-energy, it suffices to treat the effective dissipation at the mean-field level. The analytic details of our implementation of the NCA are reported in the Supplementary Note 2. The crucial feature of this extended NCA is that the resulting approximation is conserving[15,84,96–100]—namely, the approximate dynamics respect the relevant conservation laws—a necessary feature for real-time simulations. The NCA is known to have a few shortcomings, such as violating the Fermi liquid relations at zero temperature and introducing spurious features in some impurity regimes[101–104]. However, it is the simplest, conserving diagrammatic method that is able to reveal the emergence of the Kondo resonance in the relevant regime $\varepsilon_d < \mu$, and it is capable of reproducing observables like the conductance through the dot with good accuracy. Since we are more interested in finding typical (qualitative) signatures of Kondo physics rather than providing a quantitatively accurate analysis of a realistic model, we shall not attempt to overcome the limitations on the NCA in the present work.

Before describing the main results, we wish to remark that our dissipative model leads to an effective AIM physics only in the sense that a local steady state forms around the dot, as it is common with dissipative impurities[55–58,70,105–112]. Mathematically, the local steady state corresponds to the limit $\lim_{t\to\infty}\lim_{\Omega\to\infty}\rho(t)$, where $\Omega$ is the volume of the leads. The two limits do not commute: For finite leads, the system will eventually reach the true stationary manifold, which is spanned by Dicke states[113] (see also the Supplementary Note 4), including the vacuum state. In the present context, the Dicke states are best understood as eigenstates of the quadratic Hamiltonian $H$ that have no double occupancies (neither in the dot, nor in the leads) and thus are not affected by the losses. By the same token, they are exact (excited) eigenstates of the usual AIM, too. These states have correlations that are distinct from those associated with Kondo physics, and which Dicke states are reached at the end of the dynamics is determined by the initial (conserved) value of the spin. We will further discuss them in the last section of this paper and in the Supplementary Note 4. A related yet possible scenario is the formation of a finite size ferromagnetic bubble around the impurity site. The larger the bubble, the stronger the suppression of losses. On the other hand, larger bubbles are energetically unfavorable and their formation requires a less likely statistical fluctuation. In the NCA analysis, the possibility of ferromagnetic bubbles is not included, because it requires introducing spin symmetry breaking. We will discuss conditions for observing ferromagnetic bubbles in the companion paper[93].

We have computed the real-time dynamics of the effective model after a quench of the tunneling $\Gamma_T$, and we have analyzed its properties in the local steady state at late times. We first discuss the stationary properties, which are clearer to understand. The main signature of Kondo behavior is the presence

of the Kondo resonance in the impurity spectral function $A_\sigma(\omega) \equiv -\mathrm{Im} G_{d\sigma}^R(\omega)/\pi$, where $G_{d\sigma}^R(\omega)$ is the Fourier transform of the retarded impurity Green's function $G_{d\sigma}^R(t - t') \equiv -i\theta(t - t')\langle\{d_\sigma(t), d_\sigma^\dagger(t')\}\rangle$, computed in the local steady state. We refer the reader to the Supplementary Note 2 for the full time-dependent generalization of $A_\sigma(\omega)$, displayed in the Figure. We show it in Fig. 1b for an impurity tuned to the Kondo regime $\varepsilon_d = -2\Gamma_T$ in the presence of unbiased reservoirs. We observe the typical two-peak shape that one finds in the AIM[1,3,36]. There is a broad peak centered close to the single-particle energy $\varepsilon_d$, whose width is set by the original tunneling rate $\Gamma_T$. This peak reflects the rapid exchange of charge between the dot and the leads. The second peak is the Kondo resonance: a much narrower feature, pinned at the reservoirs' chemical potential $\mu_{R,L} = 0$, which signals the presence of a long-lived spin degree of freedom. In the $\gamma \to \infty$ limit, the width of the peak is set by the Kondo temperature $T_K$[1]. In the full model there would be also an extremely broad peak at the same energy $\omega \approx \varepsilon_d$, with a width of order $\gamma$ and a highly reduced height, corresponding to the doubly-occupied dot. The effect of a finite dissipation is to suppress and broaden the Kondo peak as a function of the ratio $\kappa_\sigma/T_K$, where $\kappa_\sigma$ is the Zeno-suppressed effective rate of particle loss from the system.

We can have an intuition on why a finite dissipation works against the formation of the Kondo peak by noticing[15] that it reflects the overlap between states of the system differing by one fermion (thus separated in energy by the chemical potential of the leads)—two ground states, for the Hamiltonian case (i.e., $\gamma \to +\infty$). With a finite effective dissipation, the system lacks a ground state, and all eigenstates (in the projected subspace) acquire a lifetime of order $\kappa_\sigma^{-1}$, leading to the broadening of the spectral features. As we can observe in Fig. 1b, this blurring occurs in a continuous fashion as a function of $\gamma^{-1}$. The Kondo resonance becomes significantly smeared out for $\kappa_\sigma$ grater than a few times $T_K$.

While the shape of the spectral function is a clear theoretical signature of the Kondo effect, its measurement through radio-frequency spectroscopy can be challenging in experiments with ultracold atoms, since it requires a sharp frequency resolution to reveal the Kondo peak. As mentioned in the introduction, a more accessible probe is provided by transport. The existence of the Kondo resonance can be assessed directly by measuring the differential conductance through the dot[15,15,79,96,97]. The latter is defined as $g(\Delta\mu, t) \equiv dI(\Delta\mu, t)/d\Delta\mu$, where $I(\Delta\mu, t) = -2^{-1}d(N_L(t) - N_R(t))/dt$ is the current flowing from the left reservoir to the right one for a given chemical potential bias $\Delta\mu$ [with $N_\alpha(t) \equiv \sum_{p\sigma}\langle c_{p\sigma\alpha}^\dagger c_{p\sigma\alpha}\rangle(t)$ the number of particles in reservoir $\alpha$]. This definition, which is the one employed in closed systems and also in experiments involving dissipation[59–63,114,115], does not distinguish between particles leaving a reservoir because of transport or because of losses. At zero temperature, the zero-bias (i.e., linear response) conductance directly measures the height of the Kondo peak[15,79]:

$$g(0, t) = g_{MW}(0, t) + \frac{\Gamma(\mu)}{\pi\gamma}n_d(t), \tag{7}$$

where $n_d(t) \equiv \sum_\sigma\langle d_\sigma^\dagger d_\sigma\rangle(t)$ is the dot population at time $t$. See the Supplementary Note 2 for the full expression for $g_{MW}(0, t)$. The first term of Eq. (7) converges at later times to the usual Meir-Wingreen formula[79,116] $g_\infty(0) = 2^{-1}\Gamma(\mu)\sum_\sigma A_\sigma(\mu)$. This term provides the dominant contribution, and—although formally equivalent to the $\gamma \to +\infty$ expression—it already includes most of the effects of the residual dissipation on the coherent part of transport. The second term is a small correction related to the effective losses, and it is positive, because the most populated lead at a higher chemical potential suffers an effective loss rate that is higher than the other, yielding a net current in the same direction as transport. Since we observed that a finite dissipation rate $\gamma$ decreases the height of the Kondo resonance, we expect that the zero-bias conductance $g(0, t)$ will be decreased as well. This behavior can be seen in the inset of Fig. 1b for the stationary state—the conductance decreases rapidly with $\gamma^{-1}$, from a value close to the maximal one, $2/h$ (with $h$ Planck's constant)[15]. The curves at lower values of $\varepsilon_d$ are rather far from this maximal conductance, because for them $T_K \ll \kappa_\sigma$

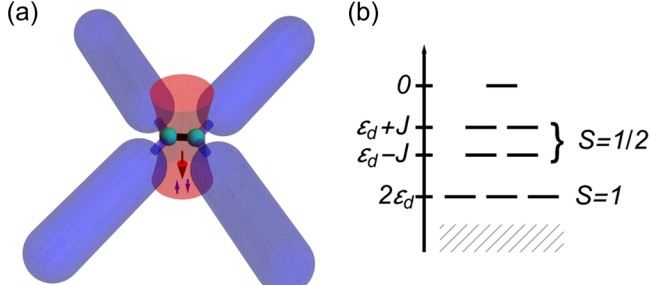

**Fig. 2 | Realizing generalized Kondo models with dissipation. a** Graphical representation of the setup in the case of two dissipative sites coupled to four leads. **b** energies of the dark states of the isolated dot with two sites. The hatched area represents the dissipated levels.

even with the largest value of $\gamma$ considered in the inset, so that the Kondo resonance is suppressed. We also notice that the various curves should not converge for $\gamma \to \infty$, since in the AIM with infinite repulsion $g_\infty(0) < 2/h$ and depends on $\varepsilon_d$[15].

As a final signature of the Kondo effect, which may prove suitable to experiments, in Fig. 1c we show the relaxation rate of the impurity magnetization $\langle\sigma^z\rangle(t) \equiv \sum_\sigma\sigma\langle d_\sigma^\dagger d_\sigma\rangle(t)$, for an initially polarized state $|\psi_0\rangle = |\uparrow\rangle_d|FS\rangle_l$, where $|FS\rangle_l$ is the ground state of the unbiased leads. We observe that, when the dot is in the Kondo regime $\varepsilon_d \lesssim -\Gamma_T$, and after an initial transient, the magnetization decays exponentially $\langle\sigma^z\rangle(t) \sim e^{-\Gamma_{decay}t}$, and that the decay rate is strongly suppressed as $\gamma$ increases. Indeed, in the limit $\gamma \to +\infty$, $\Gamma_{decay}$ is expected to be proportional to $T_K$[85,89,117], as we confirm in the Figure by showing its exponential dependence on $\varepsilon_d/\Gamma_T$. For decreasing dissipation rate $\gamma$, $\Gamma_{decay}$ converges to $\kappa_\sigma$, which is independent of $\varepsilon_d$ (but still much smaller than $\Gamma_T$). The Supplementary Note 3 provides further data on the impurity spin decay.

## Higher spin models

We have shown how local two-body losses can be used to realize a prototypical strongly correlated model with ultracold atoms, and how the resulting Kondo physics competes with the residual dissipation. We can obtain more exotic Hamiltonians if the dissipation involves more than one site, as depicted in Fig. 2a for the case of two sites and four leads. For concreteness, we take the dot to be a linear chain of $\ell_d$ sites with nearest-neighbor hopping $J$ and onsite energies $\varepsilon_d$, with open boundary conditions.

As in the case of a single dot site, we want to confine the dynamics of the system to the dark subspace of the dot, so that a weak coupling to the leads will induce only a small residual dissipation. More details on the construction can be found in the Supplementary Note 4 and in ref. 93. Figure 2b depicts the dark subspace for two impurity sites. The (strong) rotational invariance of the Lindblad equation for the isolated dot sites allows to build the dark states as simply the Dicke states[113,118,119] associated with $H_{dot}$. The resulting dark subspace is organized in multiplets of increasing spin $S = 0, \dots \ell_d/2$ and particle numbers $n = 2S$, with higher spins possessing lower energy (although there are $\binom{\ell_d}{2S}$ different multiplets with the same spin $0 < S < \ell_d/2$ but different energies, where $\binom{\cdot}{\cdot}$ is the binomial coefficient). We notice that this procedure is completely general, in the sense that it does not rely on any particular structure of the dot region, except for spin rotational invariance. In this regard, our treatment is similar to that of the Hubbard Hamiltonian with infinite repulsion[120–122].

In the setup with two sites, there is an extra complication with respect to the single-site case: the slowest-decaying eigenstate of the Lindbladian has a decay rate $\gamma_1$ which is nonmonotonic in $\gamma$, with a maximum $\gamma_1^*$ at $\gamma = 8J$ followed by a slow decrease $\gamma_1 \sim J^2/\gamma$. In contrast, all other bright states have decay rates of the order of $\gamma$. So, to keep the occupation of all bright states suppressed by dissipation, we need to have a large hopping $J \sim \gamma$, and hence a large dot energy $\varepsilon_d \sim \gamma$ to maintain the spin 1 states at energy lower than the spin 1/2 ones, $2\varepsilon_d < \varepsilon_d - J$. The nonmonotonic behavior of the smallest decay

rate, and the consequent need for fine tuning is likely to be generic for higher $\ell_d$.

If we introduce back the coupling to the leads, the latter will mediate transitions between multiplets with neighboring values of $S$, and we obtain a model that belongs to the family of the ionic models[1,36,41], that have been extensively studied in the context of magnetic impurities in metals. These models can be mapped on higher-spin Kondo models by the usual Schrieffer-Wolff transformation[1,3,11]. In the case of two sites, the lowest-energy multiplet would act like a spin 1 impurity.

While two-body losses lead naturally to higher-spin generalizations of the Kondo model, the systems considered in this work are all single-channel, in the sense that the particle exchange between the reservoirs makes them behave like a single one at thermodynamic equilibrium[1,37–39]. Nevertheless, the experimental control over the geometry and couplings of the ultra-cold atomic setups may be employed to achieve the necessary conditions for a truly multi-channel Kondo model, following the various approaches present in the literature[42–44,123].

## Data availability
The data used to generate the plots are available from the corresponding author upon reasonable request.

## Code availability
The codes employed for the numerical computations performed in this work are available from the corresponding author upon reasonable request.

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

## Acknowledgements

We acknowledge useful discussions with D. Sels, A. Gomez Salvador, R. Andrei, M. Kiselev. M.S. and J.M. have been supported by the DFG through the grant HADEQUAM-MA7003/3-1. Y.Q. and E.D. acknowledge support by the SNSF project 200021_212899, the NCCR SPIN, a National Centre of Competence in Research, funded by the Swiss National Science Foundation (grant number 225153), the Swiss State Secretariat for Education, Research and Innovation (contract number UeM019-1). E.D. also acknowledges support from the ARO grant number W911NF-20-1-0163. T.E., Y.Q., and E.D. acknowledge funding by the ETH grant. J.M. acknowledges the Pauli Center for hospitality. We gratefully acknowledge the computing time granted through the project "DysQCorr" on the Mogon II supercomputer of the Johannes Gutenberg University Mainz (hpc.uni-mainz.de), which is a member of the AHRP (Alliance for High Performance Computing in Rhineland Palatinate, www.ahrp.info), and the Gauss Alliance e.V.

## Author contributions

M.S. derived the mathematical theory and performed the numerical computations. E.D., J.M., and S.G. formulated the main physical model. Y.Q. and T.E., along with all the other authors, contributed to discussing the results and to the writing of the manuscript.

## Competing interests

The authors declare no competing interests.
