## [Transparent Peer Review file · Communications Physics]

Dissipative realization of Kondo models

Corresponding Author: Dr Yi-Fan Qu

Version 0:

Reviewer comments:

Reviewer #1

(Remarks to the Author)

The manuscript by Stefanini et al discusses a realization of the Anderson impurity model in the strong coupling limit through localized two-body losses. At low energies, this model features Kondo physics. The authors argue that this route to Kondo physics generalizes to more complex dynamic impurity systems.

I have the following comments and questions:

The authors discuss the competition between incoherent losses and coherent Kondo scattering. The authors say the residual loss rate is "proportional to the total number of fermions in the reservoirs." In this case, i.e., leads with a finite number of electrons, the universal features of the Kondo effect may be absent in such a Kondo box and spectral features will depend on the mean level spacing, see Phys. Rev. Lett. 82, 2143 (1999).

My second issue concerns the use of the NCA. The NCA is not exceptionally reliable, and it misses out on several physically relevant processes in the Kondo singlet formation. In particular, the NCA would predict a Kondo resonance in the so-called empty orbital regime where the occupation of the dissipative site is exponentially suppressed, see, e.g., the discussion in Journal of Low-Temperature Physics 126, 1233–1249 (2002) and Phys. Rev. B 70, 165102 (2004).

Concerning the reliability of approximations, Figure 1, Middle panel shows in its inset the zero-bias conductance through the dissipative site. In the Kondo regime, this quantity should approach its universal value of unitary conductance for temperatures well below the Kondo temperature. Judging from the curves in the inset, it seems unlikely to me that the top and lowest curve will approach the same value for $\gamma \rightarrow \infty$.

The generating functional for the extended NCA shown in Figure 1 of the supplemental material contains diagrams with more than one auxiliary particle loop. Yet, according to the discussion in the supplemental material, the standard pseudoparticle projection of enforcing the constraint $Q=1$ is employed. This would imply that the diagrams with more than exactly one pseudoparticle loop are suppressed by an exponential factor in the Lagrange multiplier and inverse temperature, see Phys. Rev. B 70, 165102 (2004).

The work is interesting and deserves publication in a suitable journal, but the issues/questions mentioned above keep me from recommending publication in Communications Physics at the present stage

Reviewer #2

(Remarks to the Author)

The authors consider a Fermi-gas on a one-dimensional chain. On one special site the on-site potential and tunneling matrix elements can be controlled and is subject to a adjustable dissipative term that removes two fermions if the site is doubly occupied and converts the simple resonant level model into a infinity U single impurity Anderson model in the limit of infinitely strong dissipation.

The authors argue that at large but finite dissipation, its effect can be modelled by the simultaneous removal of an on-site fermion and the neighbouring Fermion in the lead with opposite spin, described by the effective Kraus operator term L_{eff} stated in Eq (5).

The authors apply a non-equilibrium NCA as pioneered by Wingreen and Meir, Ref. [99] and treat the effective Lindblad term at finite dissipation in a Hartree approximation yielding an additional self-energy contribution to the pseudo Fermion self-energy. This theory is an expansion in $1/N$ around the atomic limit of the decoupled site and assumes thermalized leads.

The main result appears to be the on-site spectral function depicted for $T=0$ in the central panel of Fig 1. The authors nicely demonstrate the emerging resonance close to the Fermi energy for $\kappa=0$, i.e. infinitely large on-site dissipation, as expected for a SIAM in the limit $U \rightarrow \infty$. Upon increasing κ , the Kondo resonance is destroyed and not visible for $\kappa/T_K = 48.2$. The charge excitation has roughly the width $\Gamma_T = 2\Gamma_0$ where Γ_0 is the broadening usually used in the literature.

Overall I like the presented idea of dynamically created reduction of the double occupancy when increasing a dissipative term. It is original and worth a presentation. However, the paper does not explore and discuss the validity range of their $1/\gamma$ expansion. They do not discuss the known limitations of the NCA in the context of their model. Also a statement about the strategy in the numerical implementation is missing in the main text, although I agree that details can be laid out in the supplement.

I cannot recommend the paper for publication in the present form before the major issues have not been addressed.

Some major issues that should be addressed:

1)

The major assumption of the model is (a) the leads are thermalized and (b) non-interacting. How realistic are these assumptions in trapped atomic gases? Isn't a Luttinger liquid description more adequate for the leads due to the hard core repulsion?

2) On page 2, lines 111-116, last paragraph before Sec II, the authors write

"Since there is no a priori limit on the number of reservoirs coupled to the impurity sites, our proposal would pave the way to the experimental study of generalized higher-spin, many-flavor Kondo models in ultracold atoms."

The statement does not hold in general, since a multi-reservoir generalization of Fig 1 only a single effective binding orbital $C_{\sigma} = \sum_{\alpha} V_{\alpha} c_{\alpha\sigma}$ couples to the d-orbitals independent of the number of point as pointed out by Glazmann leads to a single lead problem as pointed out by Glazmann, Lee and others already 1988. Only a non-equilibrium bias might be able to change that. In general one needs a proper symmetry, which might be dynamically generated, see Oreg, Goldhaber-Gordon PRL 2003.

The authors should reword this paragraph to be more precise and accommodate the well established literature in the field.

3)

On page 3, left column, last paragraph, the Kondo scale or Kondo temperature is introduced with the statement "... governing the long-time behavior of the unitary dynamics generated by Heff,..."

This statement does not hold. The Kondo temperature is defined as a universal energy scale governing the crossover from the local moment fixed point to the strong coupling (SC) fixed point (See Anderson 1970 or Wilson RMP 1975). It defines a universal energy scale for charge and spin excitations around SC fixed point, and therefore, is an equilibrium temperature scale.

It was pointed out by Anders and Schiller (PRL 2005) that in non-equilibrium it depends on the initial condition whether TK governs the long-time behavior or not. The common belief that TK governs the long-time behavior of the unitary dynamics only holds in equilibrium or in the linear response regime.

TK is primarily a thermodynamic scale that governs the temperature evolution of the thermodynamic properties such as the spin susceptibility, the specific heat etc, as well as the low-lying charge excitation in the equilibrium spectral functions.

4)

The Kondo scale can be obtained in different ways, see Wilson RMP 1975. The most reliable way is to define it numerically by using a universality criteria: re-scaled physical properties collapse on universal functions.

The use of TK in Fig 1 is unclear. If the author uses the analytical estimate for TK stated in line 204 below Eq (5) they should state that clearly

in the figure caption.

5)

NCA limitations:

While the results appear to be convincing on first glance, the spectral functions have a several flaws caused by the approximation and the implementation of the equations that are not addressed in the text.

At $T=0$ and $\kappa=0$, we expect a sharp cusplike spectral function in NCA, from the analytic solution by Mueller-Hartmann, Z.Phys 1984. The authors cite paper by Costi et al from 1996 published 12 years later than the seminal paper by Mueller-Hartmann who analytically derived exact expressions on the spurious effects in NCA, as the authors call it. The major flaw of the NCA at low temperature is the violation of Fermi-liquid properties close to the Fermi energy, leading to the wrong scattering phase in violation of Friedel's sum rule and, as a consequence, a violation of the expected exact density of states at the Fermi energy.

The cusplike spectrum is not seen in Fig 1 which is suppose to be a $T=0$ result! The authors apparently solve the equations in the time domain with a finite time propagation that imposes a cutoff of the algebraic power law due to their numerical implementation.

It is not clear to me why they are not targeting the steady state directly, see Ref. [99], since they treat the effective dissipation only in an Hartree approximation.

6)

Upper bound for κ :

Upon decreasing γ of the original model, Eqs (1) and (2), κ increases, and the dissipation becomes weaker so that the single particle tunneling dominates. In this case, the spectral function of the resonant level model should be recovered which has a spectral width of $\Gamma_0 = \Gamma_T/2$ at the charge peak at ϵ_d . Since the author restricted themselves to the U to ∞ subspace, the broadening of the charge peak remains artificially at Γ_T although their physical model, Eq (2) should yield a reduction of the width since there is no blocking effect any more.

Since this essential feature of the original dynamics stated in Eq (1) is absent due to the restriction of the local Hilbert space, the authors should discuss this and the validity of their approximation for finite κ more thoroughly in the paper.

Ignoring the Kondo effect for a moment, the authors could, for instance, treat the local four state site by a simple rate equation, which includes the hopping on and off the thermal leads by single fermion Kraus operators ensuring detailed balance in absence of the dissipative term. In this limit, the exact resonant level model spectral function is recovered. Adding the dissipative jump operator L is straight forward and allows to explore the steady state doubly occupancy of model Eq (1) as function of increasing γ , as well as a calculation of the d-spectral function. Although the Kondo effect is absent in this crude approximation, one can learn something on the validity of the Hilbert-space reduction as employed in NCA which would give some reasonable upper bound for κ .

I doubt that the presented approximation holds for some of larger values of κ/T_K used in the central panel in Fig 1 when simulating the original model, Eq (1).

7)

Understanding the disappearance of the Kondo resonance:

On page 4, right column, paragraph starting at line 311, the authors write "We can have an intuition on why a finite dissipation works against the formation of the Kondo peak....."

Essentially, I believe, they only need to inspect the analytical properties of their NCA equations stated in their supplement. After applying a mean-field approximation for their Lindblad effective dissipation term, Eq (S23) in the supplement, any memory of the dissipation is gone and it is replaced by a delta-Function in time.

Therefore I do not fully understand why the authors apply a non-equilibrium Keldysh approach for $\mu=0$ in the time domain and not following Ref [99]

by targeting the steady state directly. Since all transients must have decayed, all steady state GFs are only dependent on the relative time

since the density operator must be stationary as well. Then, the Lindblad term becomes are frequency independent complex relaxation rate after Fourier transformation. The behaviour as depicted in Fig 1, central panel, can be immediately understood from the fact that the physical spectral function is

a convolution of the ionic propagators: the Kondo resonance emerges at low temperatures due to the threshold behaviour of the ionic (or slave fermion/boson) propagators, see analytic result by Mueller-Hartmann 1984. This threshold of the fermionic propagator is destroyed by the finite lifetime, and by self-energy correction also in the boson propagator. Consequently the Kondo resonance is suppressed in this restricted space.

Therefore, I believe, the depicted spectral evolution is correct within the restricted Hilbert space and can be understood in very simple terms with the Hartree approximation presented in the appendix.

8)

On page 4, left column, the authors write "The true stationary states in our model are fully ferromagnetic Dicke states [84, 112] that extend to the leads."

I am bit lost here. Already in a conventional Kondo equilibrium problem, there is an extend spin correlation between the local spin and the leads. In fact, the Kondo screening which is the real hallmark of the Kondo effect requires an entanglement between environment (leads) and the local spin. My impression is that referring to ferromagnetic Dicke is just a rewording of Kondo screening. One has to be careful however, spin-fluctuations play an important role and according to the Mermin-Wagner theory a symmetry broken spin imbalanced phase will be destroyed by quantum fluctuations: a Kondo singlet is a superposition of two equal contributions.

Here it is necessary to elaborate more on the difference between the conventional Kondo effect and the pseudo-Kondo model where local correlations are generated by dynamical losses and might lead to a different nature of a steady state: what is the difference between the Kondo entanglement and the Dicke states. It is not helpful to readers or referees to cite a paper [95] which is vaporware at the time of the submission.

Adding a finite symmetry breaking magnetic field to the NCA is technically not problem. However, it is well known that this leads to additional spurious results within the NCA. They are well understood and the reason why people use other methods in this case.

Minor issue:

In the supplement, the application of the Liouvillian superoperator onto the density matrix component ρ_{dd} in Eq (S3b) is incorrect. The contribution from the anticommutator leads to

$$\mathcal{L}_0 \rho_{dd} = \gamma (\rho_{00} - \rho_{dd})$$

since the trace of the density operator needs to be conserved at all times. The second term stems from the anticommutator in the Lindblad term.

Version 1:

Reviewer comments:

Reviewer #1

(Remarks to the Author)

I appreciate the authors' efforts in addressing my previous concerns and correcting the identified errors. I continue to support publication in Communications Physics. However, I have a few remaining points regarding their responses.

In my initial report, I pointed out that the NCA is not without problems. I am happy to read that the authors are very well aware of these but employ the NCA "for its main benefit, namely the ability to detect the Kondo resonance in the regime ... where it is expected". I would have thought that when probing for Kondo physics, it is more pertinent to ensure that the NCA does not produce a spurious Kondo resonance. Unfortunately, the NCA is known to produce a resonance even in the regime of the infinite-U Anderson model where the occupancy is tiny and none is expected as the strong correlation aspect of suppressed double occupancy becomes immaterial. The authors state that in the present setup, "there is a sizable probability that the dot is empty". I am not sure, if 'sizable' in this context means 99%, 90% or rather 85%, but this is the regime where the NCA may incorrectly predict a Kondo resonance. This seems an important issue to address.

The response to point 3 is intriguing. I am unaware of the Langreth relation and would have guessed that such an explicit relation can only hold in equilibrium at zero temperature. But of course, I accept the point of the authors that the occupation deviates considerably from $n_\sigma \approx 1/2$. Given that the Kondo model is particle-hole symmetric, I expect G to be symmetric around that point.

The Langreth relation is not symmetric around $n_\sigma = 1/2$.

The response to point 4 is also interesting. Again, my naive take is that the Keldysh technique can be applied in equilibrium and then will be equivalent to the Matsubara technique. I read your reply as saying that in that case (for the NCA) there will

be a difference in the diagrammatic structure of the approximation. I always thought that a rigorous projection of pseudoparticle propagators leads to objects that are very close to lesser and larger functions (instead of retarded and advanced) due to the removal of back propagation in the projection.

Reviewer #2

(Remarks to the Author)

The authors satisfactory answered all questions raised by both reviewers and edited their manuscript accordingly. They added some clarifications and corrected the typo in one of their equations in the supplement.

I recommend publishing the paper after these revisions.

Version 2:

Reviewer comments:

Reviewer #1

(Remarks to the Author)

The authors have responded to my feedback and revised the manuscript correspondingly. Although my initial concerns persist to some extent, I believe the paper meets the threshold for publication.

Rebuttal: comments to Reviewers
for the manuscript
“Dissipative realization of Kondo models”

Martino Stefanini, Yi-Fan Qu, Tilman Esslinger,
Sarang Gopalakrishnan, Eugene Demler, and Jamir Marino

January 2025

1 Reply to Referee 1

We thank the Referee for the comments. In the following, we will address the main issues raised. The original comments have been copied in *italics*.

1. *The authors discuss the competition between incoherent losses and coherent Kondo scattering. The authors say the residual loss rate is “proportional to the total number of fermions in the reservoirs.” In this case, i.e., leads with a finite number of electrons, the universal features of the Kondo effect may be absent in such a Kondo box and spectral features will depend on the mean level spacing, see Phys. Rev. Lett. 82, 2143 (1999).*

The reference to the total number of fermions is a typo. The correct statement is that the residual rate is proportional to the total density of fermions—indeed, Eq. (6) is intensive. This point has been corrected in the manuscript.

2. *My second issue concerns the use of the NCA. The NCA is not exceptionally reliable, and it misses out on several physically relevant processes in the Kondo singlet formation. In particular, the NCA would predict a Kondo resonance in the so-called empty orbital regime where the occupation of the dissipative site is exponentially suppressed, see, e.g., the discussion in Journal of Low-Temperature Physics 126, 1233-1249 (2002) and Phys. Rev. B 70, 165102 (2004).*

We are aware of the limitations of the NCA. Our point of view is that we are employing it for its main benefit, namely the ability to detect the Kondo resonance in the regime $\varepsilon_d < \mu$ where it is expected. Our aim is to show that there is the Kondo effect in a simplified model, while an accurate prediction of the properties of the model in all possible regimes, that would require to overcome the limitations of the NCA, is beyond the scope of the work. Besides, the NCA has the property of being a conserving

approximation, which is crucial for obtaining physically sensible results in a dynamical setting, and it is the simplest approximation of this kind. We have emphasized our point of view in the main text, where the NCA is introduced.

3. *Concerning the reliability of approximations, Figure 1, Middle panel shows in its inset the zero-bias conductance through the dissipative site. In the Kondo regime, this quantity should approach its universal value of unitary conductance for temperatures well below the Kondo temperature. Judging from the curves in the inset, it seems unlikely to me that the top and lowest curve will approach the same value for $\gamma \rightarrow \infty$.*

Unlike in the Kondo model, in the AIM with infinite repulsion the conductance does not necessarily reach its unitary value: see, for instance, Fig. 6 in [1]. This can be seen from Langreth's relation $G = 2e^2/h \sin \pi \langle n_\sigma \rangle$ —while $\langle n_\sigma \rangle$ is fixed at 1/2 in the Kondo model, yielding the maximal conductance, in the infinite-U AIM $\langle n_\sigma \rangle$ depends on $\varepsilon_d - \mu$ and is generally lower than 1/2 (since there is a sizable probability that the dot is empty). The NCA does not satisfy Langreth's relation, but the conclusions are confirmed by previous studies such as [1] quoted before. So, in general the various curves in the inset to the central figure need not converge to the same value. Still, one may possibly expect a larger value of conductance for the lower curves, i.e. the ones for a deeper impurity state. Our explanation for their small values is that for those curves T_K is so low that it is still much smaller than the effective decay rate κ_σ , so that the Kondo resonance is still largely suppressed, like the yellow curve in the central panel. We have added a footnote to explain this point.

4. *The generating functional for the extended NCA shown in Figure 1 of the supplemental material contains diagrams with more than one auxiliary particle loop. Yet, according to the discussion in the supplemental material, the standard pseudoparticle projection of enforcing the constraint $Q=1$ is employed. This would imply that the diagrams with more than exactly one pseudoparticle loop are suppressed by an exponential factor in the Lagrange multiplier and inverse temperature, see *Phys. Rev. B* 70, 165102 (2004).*

We are following the projection procedure of Refs. [1–3], which is directly formulated in the Keldysh formalism. In this procedure, the equivalent of the exponential factor is attached only to the lesser Green's function of auxiliary particles, and the projection has to be done by introducing the lesser function explicitly (e.g. if $B^>$ appears, this must be converted to $B^R + B^<$ to apply the projection). Then, in this formalism the number of auxiliary particle loops is limited to zero or one, depending on the self energy that is considered. Retarded Green's functions have no suppression factor, hence the corresponding retarded self-energies must have no auxiliary particle loops (or, more generally, no lesser functions of auxiliary particles). Lesser Green's functions have one suppression factor, so their

lesser self-energies are allowed to have one loop (i.e. one lesser function of auxiliary particles). Indeed, the dissipative part of the retarded self-energies feature no auxiliary particle loops after the projection (Eqs. (23) in the Supplemental): the auxiliary fermions' self-energy ends up with just a loop of the leads fermions (which carries no suppression factor, since it is the bare one), while the auxiliary boson self-energy vanishes because it should contain a loop. The lesser self-energies, instead (Eqs. (26)), contain just one loop (although the fermionic one vanishes even before the projection). Figure 1 in the Supplemental shows the diagrams before the projection. We have clarified this point in the caption. We also notice that if the mean-field diagrams for the dissipative parts were blindly discarded, this would imply that the leading effect of the dissipation would occur at the next order in $1/\gamma$, which would be physically unexpected and, indeed, incorrect. The total rate of particle loss has the scaling $I_{\text{loss}} \sim \kappa_{\sigma} \sim 1/\gamma$, which is confirmed by small-scale exact diagonalization and by the variational results of the companion paper. If we discarded the mean-field diagram, I_{loss} would decay at least as $1/\gamma^2$.

2 Reply to Referee 2

In the following, we answer to the issues raised by the Referee, who we thank for the thorough analysis of the manuscript. The original comments have been copied in *italics*.

1. *The major assumption of the model is (a) the leads are thermalized and (b) non-interacting. How realistic are these assumptions in trapped atomic gases? Isn't a Luttinger liquid description more adequate for the leads due to the hard core repulsion?*

The two assumptions of initial equilibrium and lack of interactions are indeed quite realistic in the field of ultra-cold atoms. Atomic clouds can be prepared in the degenerate regime with temperatures down to about 10% of the Fermi energy. And these systems are quite dilute, so that interactions are due to Van der Waals forces rather than the hard-core repulsion. Thus, the s-wave scattering length is all that is needed to know to characterize scattering events in this regime. Moreover, the scattering length can be tuned via external magnetic fields by exploiting the presence of Feshbach resonances, so that it is essentially a free parameter. Thus, noninteracting leads can be obtained in experiments. This is especially true in the case of Lithium-6 atoms, that is more relevant to our model. These properties of ultra-cold atom systems are the subject of many reviews; for the present transport setup, we can recommend Ref. [4], written by one of the authors and cited in the paper.

If the concern of the Referee is that the lack of interactions may prevent thermalization, we can provide reassurance that this is not the case, since the atomic cloud is not an isolated system during its preparation,

and even weak interactions can ensure thermalization. The reaching of thermal equilibrium is usually not considered an issue in the preparation of ultracold atomic systems [5–8].

Finally, in realistic systems the leads are three-dimensional. For this reason, in the presence of weak interactions the appropriate description would be in terms of Fermi liquids rather than Luttinger liquids, and this justifies the noninteracting Hamiltonian used in the text, exactly as it happens in solid state. In this sense, adding weak interactions in the leads would complicate the theoretical treatment without adding much to the physics. And, in our opinion, the results that we have found are much more stark if all correlations can be attributed to dissipation. We have added a footnote clarifying these issues, and we modified the schematic representation of the system in both figures.

2. *On page 2, lines 111-116, last paragraph before Sec II, the authors write "Since there is no a priori limit on the number of reservoirs coupled to the impurity sites, our proposal would pave the way to the experimental study of generalized higher-spin, many-flavor Kondo models in ultracold atoms." The statement does not hold in general, since a multi-reservoir generalization of Fig1 only a single effective binding orbital $C_\sigma = N \sum_{\alpha p} V_{p\alpha} c_{p\alpha\sigma}$ couples to the d-orbitals independent of the number of point as pointed out by Glazmann leads to a single lead problem as pointed out by Glazmann, Lee and others already 1988. Only a non-equilibrium bias might be able to change that. In general one needs a proper symmetry, which might be dynamically generated, see Oreg, Goldhaber-Gordon PRL 2003. The authors should reword this paragraph to be more precise and accommodate the well established literature in the field.*

We are aware that simply adding reservoirs does not lead to a multi-channel Kondo model. What we meant is that the cold-atom setup is sufficiently flexible to be generalized in that direction, because of the experimental control over the various couplings (while the use of dissipation does not offer any intrinsic advantage). We thank the Referee for pointing out the ambiguous phrasing of the manuscript. We have re-written the paragraphs at lines 111-116 and at the end to avoid the ambiguity.

3. *On page 3, left column, last paragraph, the Kondo scale or Kondo temperature is introduced with the statement "... governing the long-time behavior of the unitary dynamics generated by Heff,..." This statement does not hold. [...] T_K is primarily a thermodynamic scale govern the temperature evolution of the thermodynamic properties such as the spin susceptibility, the specific heat etc, as well as the low-lying charge excitation in the equilibrium spectral functions.*

We agree that T_K emerges primarily as a thermodynamical quantity. However, there is a substantial amount of literature that points out that it plays an important role even in dynamical situations, such as [9–15], and our own computations for the spin decay seem to confirm this role. Besides, the

role of T_K in stationary but non-equilibrium situation is well-established [1, 16–19] (a particularly comprehensive list of papers can be found in the bibliography of [15]), and the latter would surely apply to the local stationary state of our model. As far as we can see, we are not exciting our models to energies such that T_K is not sufficient anymore. Indeed, our setup is consistent with the Anders-Schiller results, since the spin relaxation is measured starting from an uncoupled spin, for which both we and Anders-Schiller find universal behavior in $T_K t$. We have clarified this point with a footnote to the text.

4. [...] *The use of TK in Fig 1 is unclear. If the author use the analytical estimate for TK stated in line 204 below Eq (5) they should state that clearly in the figure caption.*

There is only one expression for T_K mentioned in the whole paper, so we had taken as implied that we stick to that definition. From the Referee’s comment we acknowledge that this implication might not be obvious to readers, so we have modified the caption for Fig. 1 accordingly.

5. *NCA limitations:*

While the results appear to be convincing on first glance, the spectral functions have a several flaws caused by the approximation and the implementation of the equations that are not addressed in the text.

At $T=0$ and $\kappa = 0$, we expect a sharp cusplike spectral function in NCA, from the analytic solution by Mueller-Hartmann, Z.Phys 1984. The authors cite paper by Costi et al from 1996 published 12 years later than the seminal paper by Mueller-Hartmann who analytically derived exact expressions on the spurious effects in NCA, as the authors call it. The major flaw of the NCA at low temperature is the violation of Fermi-liquid properties close to the Fermi energy, leading to the wrong scattering phase in violation of Friedel’s sum rule and, as a consequence, a violation of the expected exact density of states at the Fermi energy.

The cusplike spectrum is not seen in Fig 1 which is suppose to be a $T = 0$ result! The authors apparently solve the equations in the time domain with a finite time propagation that imposes a cutoff of the algebraic power law due to their numerical implementation. It is not clear to me why they are not targeting the steady state directly, see Ref. [99], since they treat the effective dissipation only in an Hartree approximation.

The dynamical perspective on the model allows us to be closer in spirit to experiments, since in general the latter operate far from the stationary regime. Thus, the spectral functions that we compute would be more similar to the ones that radio-frequency (rf) spectroscopy would reveal (in a very idealized scenario: rf spectroscopy could not probably observe the Kondo peak, due to the limited frequency resolution). The closeness to experiments is also the reason for considering the spin decay rate as a proxy for revealing the Kondo effect, because it is a dynamical effect on an accessible quantity.

The apparent absence of the cusp at zero frequency can be attributed to a few concomitant factors. First, the amplitude of the cusp is rather small in the Kondo regime, and second, it is presumably overshadowed by the Kondo peak, which is essentially at $\omega = 0$ as well. Distinguishing them would require a finer frequency resolution. This could be more easily done in the local stationary state, but we do not see the usefulness of this calculation, since it would highlight a pathological feature of the approximation that is not present in reality, while adding nothing to the physical picture.

We are aware of the Müller-Hartmann solution at zero temperature, and we even adapted his calculation to the present model. However, we chose not to include it in our paper since it is essentially of scarce utility in the present context. Indeed, it allows to compute semi-analytically only the retarded Green's functions pseudo-particles. While at equilibrium the fluctuation-dissipation theorem (FDT) allows to derive the lesser Green's functions from the retarded ones, and thus to reconstruct the full impurity Green's function, this is no longer possible in our dissipative, out of equilibrium situation (not even in the local stationary state), since the FDT ceases to be valid.

In general, our paper is meant to be a proof of principle of the emergence of Kondo physics from a purely dissipative situation. In this spirit, we employ the NCA as the simplest approach that is able to reveal Kondo physics in the parameter regimes in which we expect it to be present. The fact that it is also a conserving approximation makes it even more relevant to the present out-of-equilibrium scenario. The various known spurious effects that it introduces are of no particular concern to us, since it is not our goal to produce quantitative predictions to be compared to experiments (this would require, e.g., a better description of the reservoirs, beyond the wide band limit). Essentially, the impurity spectral function serves only as a litmus test for the emergence of Kondo physics, while it is not accessible experimentally.

6. *Upper bound for κ :*

Upon decreasing γ of the original model, Eqs (1) and (2), κ increases, and the dissipation becomes weaker so that the single particle tunneling dominates. In this case, the spectral function of the resonant level model should be recovered which has a spectral width of $\Gamma_0 = \Gamma_T/2$ at the charge peak at ϵ_d . Since the author restricted themselves to the $U \rightarrow \infty$ subspace, the broadening of the charge peak remains artificially at Γ_T although their physical model, Eq (2) should yield a reduction of the width since there is no blocking effect any more.

The persistent broadening of the charge peak can be understood by noticing that even the lowest value of $\gamma = W$ that we considered (hence, the largest κ) is still 100 times larger than Γ_T , so that the system is still in the dissipation-dominated regime, with essentially no double occupancies (later, we will estimate $\langle n_{d\uparrow}n_{d\downarrow} \rangle \sim 10^{-2}$). To see the behavior that the

Referee is suggesting one should decrease γ below the bandwidth of the bath (and probably down to about Γ_T), which is outside of the regime in which the effective model is valid. We notice that, in the full theory, we expect to see always one charge peak, since both processes $|0\rangle \rightarrow |\sigma\rangle$ and $|\sigma\rangle \rightarrow |d\rangle$ cost the same energy ε_d (neglecting possible dissipative renormalizations of the single-particle impurity level). Upon increasing the loss rate γ beyond Γ_T , the former process would keep a width of the order of Γ_T , while the $|\sigma\rangle \rightarrow |d\rangle$ process should acquire a large width of the order of $\Gamma_T + \gamma \approx \gamma$. In other words, we expect a peak-and-shoulder structure to emerge. It is unlikely that we may be witnessing the emergence of such a structure in the effective model when decreasing γ . The slight increase of the width of the charge peak in Fig. 1 in the manuscript is more easily interpreted as a redistribution of the weight under the Kondo peak as the latter gets suppressed (the total area under the spectral function is $1 - n_{d\bar{\sigma}}$ in the projected subspace, but the impurity population decreases by less than 10% in the whole interval of γ that we considered).

Since this essential feature of the original dynamics stated in Eq (1) is absent due to the restriction of the local Hilbert space, the authors should discuss this and the validity of their approximation for finite κ more thoroughly in the paper.

Ignoring the Kondo effect for a moment, the authors could, for instance, treat the local four state site by a simple rate equation, which includes the hopping on and off the thermal leads by single fermion Kraus operators ensuring detailed balance in absence of the dissipative term. In this limit, the exact resonant level model spectral function is recovered. Adding the dissipative jump operator L is straight forward and allows to explore the steady state doubly occupancy of model Eq (1) as function of increasing γ , as well as a calculation of the d -spectral function. Although the Kondo effect is absent in this crude approximation, one can learn something on the validity of the Hilbert-space reduction as employed in NCA which would give some reasonable upper bound for κ .

I doubt that the presented approximation holds for some of larger values of κ/T_K used in the central panel in Fig 1 when simulating the original model, Eq (1).

We have introduced a short discussion of the parametric region of validity of the effective model. However, the toy model that the Referee proposes cannot be compared directly to the effective model, since the former ceases to be valid when the latter begins to be applicable. Indeed, to approximate the effect of the leads via a Lindbladian treatment, the basic requirement would be that the leads should be considered Markovian with respect to the dynamics of the impurity site (including the two-body loss), i.e. the timescales of the internal dynamics of the reservoirs should be the shortest in the problem. In particular, this requires that γ has to be much smaller than the leads' bandwidth W (although it could be still much larger than $|\varepsilon_d|$ and Γ_T). On the other hand, the effective model rests

on the assumption that γ is the largest energy scale, even bigger than W . In other words, the dissipative part of the impurity dynamics occurs on a timescale during which the baths are frozen, i.e. non-Markovian. Indeed, the double occupancies $\delta = \langle n_{d\uparrow}n_{d\downarrow} \rangle$ obey different scaling laws in the two models: in the toy model $\delta \sim \Gamma_T/\gamma$, while in the effective model $\delta \sim W\Gamma_T/\gamma^2$ [Eq. (52) in the Supplemental].

As mentioned in the Supplemental Material, the small parameters in the derivation of the effective model are $|\varepsilon_d|/\gamma$, Γ_T/γ and W/γ . Since we always work in the regime $|\varepsilon_d| \sim \Gamma_T \ll W$, the lower bound for γ is represented by the bandwidth W . The largest value of κ in Fig. 1 corresponds to $\gamma = W$ and thus lies at the very boundary of validity of the effective model. Notice that in this case $\kappa \approx \Gamma_T$ (which is, however, much bigger than T_K), while the double occupancies are still greatly suppressed as $\delta \sim W\Gamma_T/\gamma^2 \sim \Gamma_T/W \sim 10^{-2}$ (we notice that this estimate of δ is presumably overestimating it for $\gamma \lesssim (W\Gamma_T)^{1/2}$, where the effective model ceases to be valid). By the latter estimate, the exclusion of doubly occupied states in the effective model seems to be well justified in all the plots. We have added a paragraph discussing the limits of the effective model at the end of the “Model” section.

7. *Understanding the disappearance of the Kondo resonance:*

On page 4, right column, paragraph starting at line 311, the authors write “We can have an intuition on why a finite dissipation works against the formation of the Kondo peak.....”

Essentially, I believe, they only need to inspect the analytical properties of their NCA equations stated in their supplement.[...] Therefore, I believe, the depicted spectral evolution is correct within the restricted Hilbert space and can be understood in very simple terms with the Hartree approximation presented in the appendix.

We appreciate that the Referee finds the spectral function that we have computed physically reasonable. We also find that the Referee’s explanation of the competition between residual losses and Kondo resonance is simply a more refined version of what is written in the main text. We were aware of this explanation, but we found it more technically involved, though, and it would have taken more space in the text. So, we have preferred to stick to the more hand-waving—and, possibly, more physically intuitive—discussion that is present in the paper.

8. *On page 4, left column, the authors write “The true stationary states in our model are fully ferromagnetic Dicke states [84, 112] that extend to the leads.”*

I am bit lost here. Already in a conventional Kondo equilibrium problem, there is an extend spin correlation between the local spin and the leads. In fact, the Kondo screening which is the real hallmark of the Kondo effect requires an entanglement between environment (leads) and the local spin. My impression is that referring to ferromagnetic Dicke is just a rewording

of Kondo screening. One has to be careful however, spin-fluctuations play an important role and according to the Mermin-Wagner theory a symmetry broken spin imbalanced phase will be destroyed by quantum fluctuations: a Kondo singlet is a superposition of two equal contributions.

Here it is necessary to elaborate more on the difference between the conventional Kondo effect and the pseudo-Kondo model where local correlations are generated by dynamical losses and might lead to a different nature of a steady state: what is the difference between the Kondo entanglement and the Dicke states. It is not helpful to readers or referees to cite a paper [95] which is vaporware at the time of the submission.

Adding a finite symmetry breaking magnetic field to the NCA is technically not a problem. However, it is well known that this leads to additional spurious results within the NCA. They are well understood and the reason why people use other methods in this case.

There has been a misunderstanding on the issue of stationary states, evidently caused by a lack of clarity in the original text. We have re-written that paragraph in a more detailed manner, to avoid further confusion. We have also added a remark in the Supplemental that further explains them, since they were implicit in the discussion of the setup with many dot sites.

The Dicke states are not related to the Kondo effect, and feature completely different spin correlations. They can be built by first diagonalizing the resonant level model $H = H_d + H_{\text{leads}} + H_{\text{tun}}$, forming a completely polarized eigenstate $|\uparrow \dots \uparrow\rangle$ of N_f fermions distributed among the available energy levels of H (i.e. the fermions will be delocalized between the dot and the leads), and then repeatedly applying the total spin-lowering operator $S^- = d_{\downarrow}^{\dagger} d_{\uparrow} + \sum_{p\alpha} c_{p\alpha\downarrow}^{\dagger} c_{p\alpha\uparrow}$ (followed by a proper normalization). Since H commutes with S^- , all the states generated in this way are eigenstates of H . The resulting multiplet¹ of states spans a representation of spin $S = N_f/2$ of the total spin (dot plus leads) operator, with the two fully polarized states representing the states of maximal magnetization $M = \pm N_f/2$. The vacuum $|0\rangle$ is the simplest Dicke state, with total spin 0. The important property of these states is that they are completely devoid of double occupancies², and in particular they are annihilated by the jump operator L . All of these properties guarantee that, if $|\phi\rangle$ is any Dicke state, then $\rho_D = |\phi\rangle\langle\phi|$ is a stationary state of the full model (1), and the dynamics of (1) will eventually lead the state of the dot plus leads system in the Dicke manifold. These states are evidently much different from the Kondo state, and in this sense our proposal does not simply generate an effective Kondo model/AIM for the stationary state. As mentioned in the main text, the Kondo physics emerges during the dynamics as a local

¹For any N_f which is not zero nor equal to the total number of available states, there are different multiplets for the same value of the total spin, depending on the choice of the filled levels in the initial fully polarized state.

²They can also be written as Slater determinants in which the exclusion principle is extended to both spin projections, i.e. with one *spinful* fermion per single-particle state.

steady state for distanced d within the light-cone $d < v_F t$, where v_F is the Fermi velocity of the lead(s). This local steady state is a transient phenomenon, because it is not a true stationary state. However, the crossover to the Dicke manifold takes a very long time to take place, possibly on a timescale proportional to L^3 , where L is the linear size of the leads (this estimate is extracted from [20, 21]), so by taking the limit of infinitely large leads (as it happens in the NCA) this process is completely frozen and the local, Kondo-like steady state becomes the true steady state. We would like to remark that the Dicke states are also exact eigenstates of the usual AIM, although with higher energy than the Kondo ground state.

With regards to the difference in the correlation between the ordinary AIM and the loss-induced AIM presented here, the question is definitely interesting but cannot be answered within the NCA treatment employed in this paper (it would require the calculation of higher order correlation functions, an unwieldy task out of equilibrium). We have tried to answer that question in the companion paper, which is now on arXiv, arXiv:2411.13638.

9. *Minor issue:*

In the supplement, the application of the Liouvillian superoperator onto the density matrix component $|dd\rangle$ in Eq (S3b) is incorrect. The contribution from the anticommutator leads to

$$\mathcal{L}_0|dd\rangle = \gamma(|00\rangle - |dd\rangle)$$

since the trace of the density operator needs to be conserved at all times. The second term stems from the anticommutator in the Lindblad term.

We thank the Referee for pointing out the typo, which has been corrected. The subsequent formulas imply the correct form $\mathcal{L}_0|dd\rangle = \gamma(|00\rangle - |dd\rangle)$.

References

- [1] Ned S. Wingreen and Yigal Meir. “Anderson model out of equilibrium: Noncrossing-approximation approach to transport through a quantum dot”. In: *Phys. Rev. B* 49 (16 Apr. 1994), pp. 11040–11052. DOI: [10.1103/PhysRevB.49.11040](https://doi.org/10.1103/PhysRevB.49.11040). URL: <https://link.aps.org/doi/10.1103/PhysRevB.49.11040>.
- [2] David C. Langreth and P. Nordlander. “Derivation of a master equation for charge-transfer processes in atom-surface collisions”. In: *Phys. Rev. B* 43 (4 Feb. 1991), pp. 2541–2557. DOI: [10.1103/PhysRevB.43.2541](https://doi.org/10.1103/PhysRevB.43.2541). URL: <https://link.aps.org/doi/10.1103/PhysRevB.43.2541>.
- [3] Hongxiao Shao, David C. Langreth, and Peter Nordlander. “Many-body theory for charge transfer in atom-surface collisions”. In: *Phys. Rev. B* 49 (19 May 1994), pp. 13929–13947. DOI: [10.1103/PhysRevB.49.13929](https://doi.org/10.1103/PhysRevB.49.13929). URL: <https://link.aps.org/doi/10.1103/PhysRevB.49.13929>.

- [4] Sebastian Krinner, Tilman Esslinger, and Jean-Philippe Brantut. “Two-terminal transport measurements with cold atoms”. In: *Journal of Physics: Condensed Matter* 29.34 (July 2017), p. 343003. DOI: [10.1088/1361-648X/aa74a1](https://doi.org/10.1088/1361-648X/aa74a1). URL: <https://dx.doi.org/10.1088/1361-648X/aa74a1>.
- [5] Claude N. Cohen-Tannoudji. “Nobel Lecture: Manipulating atoms with photons”. In: *Rev. Mod. Phys.* 70 (3 July 1998), pp. 707–719. DOI: [10.1103/RevModPhys.70.707](https://link.aps.org/doi/10.1103/RevModPhys.70.707). URL: <https://link.aps.org/doi/10.1103/RevModPhys.70.707>.
- [6] William D. Phillips. “Nobel Lecture: Laser cooling and trapping of neutral atoms”. In: *Rev. Mod. Phys.* 70 (3 July 1998), pp. 721–741. DOI: [10.1103/RevModPhys.70.721](https://link.aps.org/doi/10.1103/RevModPhys.70.721). URL: <https://link.aps.org/doi/10.1103/RevModPhys.70.721>.
- [7] Immanuel Bloch, Jean Dalibard, and Wilhelm Zwerger. “Many-body physics with ultracold gases”. In: *Rev. Mod. Phys.* 80 (3 July 2008), pp. 885–964. DOI: [10.1103/RevModPhys.80.885](https://link.aps.org/doi/10.1103/RevModPhys.80.885). URL: <https://link.aps.org/doi/10.1103/RevModPhys.80.885>.
- [8] Tilman Esslinger. “Fermi-Hubbard Physics with Atoms in an Optical Lattice”. In: *Annual Review of Condensed Matter Physics* 1. Volume 1, 2010 (2010), pp. 129–152. ISSN: 1947-5462. DOI: <https://doi.org/10.1146/annurev-conmatphys-070909-104059>. URL: <https://www.annualreviews.org/content/journals/10.1146/annurev-conmatphys-070909-104059>.
- [9] Peter Nordlander, Michael Pustilnik, Yigal Meir, Ned S. Wingreen, and David C. Langreth. “How Long Does It Take for the Kondo Effect to Develop?” In: *Phys. Rev. Lett.* 83 (4 July 1999), pp. 808–811. DOI: [10.1103/PhysRevLett.83.808](https://link.aps.org/doi/10.1103/PhysRevLett.83.808). URL: <https://link.aps.org/doi/10.1103/PhysRevLett.83.808>.
- [10] Martin Plihal, David C. Langreth, and Peter Nordlander. “Transient currents and universal time scales for a fully time-dependent quantum dot in the Kondo regime”. In: *Phys. Rev. B* 71 (16 Apr. 2005), p. 165321. DOI: [10.1103/PhysRevB.71.165321](https://link.aps.org/doi/10.1103/PhysRevB.71.165321). URL: <https://link.aps.org/doi/10.1103/PhysRevB.71.165321>.
- [11] Dmitry Lobaskin and Stefan Kehrein. “Crossover from nonequilibrium to equilibrium behavior in the time-dependent Kondo model”. In: *Phys. Rev. B* 71 (19 May 2005), p. 193303. DOI: [10.1103/PhysRevB.71.193303](https://link.aps.org/doi/10.1103/PhysRevB.71.193303). URL: <https://link.aps.org/doi/10.1103/PhysRevB.71.193303>.
- [12] H. T. M. Nghiem and T. A. Costi. “Time Evolution of the Kondo Resonance in Response to a Quench”. In: *Phys. Rev. Lett.* 119 (15 Oct. 2017), p. 156601. DOI: [10.1103/PhysRevLett.119.156601](https://link.aps.org/doi/10.1103/PhysRevLett.119.156601). URL: <https://link.aps.org/doi/10.1103/PhysRevLett.119.156601>.

- [13] Yuto Ashida, Tao Shi, Mari Carmen Bañuls, J. Ignacio Cirac, and Eugene Demler. “Solving Quantum Impurity Problems in and out of Equilibrium with the Variational Approach”. In: *Phys. Rev. Lett.* 121 (2 July 2018), p. 026805. DOI: [10.1103/PhysRevLett.121.026805](https://doi.org/10.1103/PhysRevLett.121.026805). URL: <https://link.aps.org/doi/10.1103/PhysRevLett.121.026805>.
- [14] Yuto Ashida, Tao Shi, Mari Carmen Bañuls, J. Ignacio Cirac, and Eugene Demler. “Variational principle for quantum impurity systems in and out of equilibrium: Application to Kondo problems”. In: *Phys. Rev. B* 98 (2 July 2018), p. 024103. DOI: [10.1103/PhysRevB.98.024103](https://doi.org/10.1103/PhysRevB.98.024103). URL: <https://link.aps.org/doi/10.1103/PhysRevB.98.024103>.
- [15] Moallison F. Cavalcante, Rodrigo G. Pereira, and Maria C. O. Aguiar. “Quench dynamics of the Kondo effect: Transport across an impurity coupled to interacting wires”. In: *Phys. Rev. B* 107 (7 Feb. 2023), p. 075110. DOI: [10.1103/PhysRevB.107.075110](https://doi.org/10.1103/PhysRevB.107.075110). URL: <https://link.aps.org/doi/10.1103/PhysRevB.107.075110>.
- [16] Tai Kai Ng and Patrick A. Lee. “On-Site Coulomb Repulsion and Resonant Tunneling”. In: *Phys. Rev. Lett.* 61 (15 Oct. 1988), pp. 1768–1771. DOI: [10.1103/PhysRevLett.61.1768](https://doi.org/10.1103/PhysRevLett.61.1768). URL: <https://link.aps.org/doi/10.1103/PhysRevLett.61.1768>.
- [17] Leonid I. Glazman and M. E. Raikh. “Resonant Kondo transparency of a barrier with quasilocal impurity states”. In: *JETP Letters* 47 (8 Apr. 1988), p. 378. URL: http://jetpletters.ru/ps/0/article_16538.shtml.
- [18] Matthias H. Hettler, Johann Kroha, and Selman Hershfield. “Nonequilibrium dynamics of the Anderson impurity model”. In: *Phys. Rev. B* 58 (9 Sept. 1998), pp. 5649–5664. DOI: [10.1103/PhysRevB.58.5649](https://doi.org/10.1103/PhysRevB.58.5649). URL: <https://link.aps.org/doi/10.1103/PhysRevB.58.5649>.
- [19] Noam Sivan and Ned S. Wingreen. “Single-impurity Anderson model out of equilibrium”. In: *Phys. Rev. B* 54 (16 Oct. 1996), pp. 11622–11629. DOI: [10.1103/PhysRevB.54.11622](https://doi.org/10.1103/PhysRevB.54.11622). URL: <https://link.aps.org/doi/10.1103/PhysRevB.54.11622>.
- [20] Masaya Nakagawa, Norio Kawakami, and Masahito Ueda. “Exact Liouvillian Spectrum of a One-Dimensional Dissipative Hubbard Model”. In: *Phys. Rev. Lett.* 126 (11 Mar. 2021), p. 110404. DOI: [10.1103/PhysRevLett.126.110404](https://doi.org/10.1103/PhysRevLett.126.110404). URL: <https://link.aps.org/doi/10.1103/PhysRevLett.126.110404>.
- [21] Hironobu Yoshida and Hosho Katsura. “Liouvillian gap and single spin-flip dynamics in the dissipative Fermi-Hubbard model”. In: *Phys. Rev. A* 107 (3 Mar. 2023), p. 033332. DOI: [10.1103/PhysRevA.107.033332](https://doi.org/10.1103/PhysRevA.107.033332). URL: <https://link.aps.org/doi/10.1103/PhysRevA.107.033332>.

Second rebuttal: comments to Reviewers
for the manuscript
“Dissipative realization of Kondo models”

Martino Stefanini, Yi-Fan Qu, Tilman Esslinger,
Sarang Gopalakrishnan, Eugene Demler, and Jamir Marino

March 2025

Reply to Referee 1

We thank the Referee for appreciating our efforts. In the following, we will address the remaining issues. The original comments have been copied in *italics*.

1. *In my initial report, I pointed out that the NCA is not without problems. I am happy to read that the authors are very well aware of these but employ the NCA “for its main benefit, namely the ability to detect the Kondo resonance in the regime ... where it is expected”. I would have thought that when probing for Kondo physics, it is more pertinent to ensure that the NCA does not produce a spurious Kondo resonance. Unfortunately, the NCA is known to produce a resonance even in the regime of the infinite- U Anderson model where the occupancy is tiny and none is expected as the strong correlation aspect of suppressed double occupancy becomes immaterial. The authors state that in the present setup, “there is a sizable probability that the dot is empty”. I am not sure, if ‘sizable’ in this context means 99%, 90% or rather 85%, but this is the regime where the NCA may incorrectly predict a Kondo resonance. This seems an important issue to address.*

There seems to be a misunderstanding here: in all parameters regimes that we consider, the impurity is never mostly empty. In general, the total occupancy is never below 0.4, which means that the empty orbital regime is never attained. Indeed, we always consider the regime $\varepsilon_d < \mu$ in which the dot would tend to fill up in the absence of dissipation, and we have verified that at the very high loss rates that we considered the occupancy actually increases with γ . In the Anderson impurity model at infinite repulsion, the regime $\varepsilon_d < \mu$ is precisely the only regime in which the Kondo resonance develops. (The only distinction is whether $\mu - \varepsilon_d$ is of order Γ_T or larger, the former being the “mixed-valence” regime with a broad resonance while the latter is the “proper” Kondo regime with n_d

tending to unity and a very narrow resonance. We have covered both regimes of the crossover.) Our effective model at large dissipation is a perturbation (albeit non-Hamiltonian) of the AIM, and our computations indicate that the perturbation does not induce drastic changes—the effective dissipation has a similar effect to finite temperature, decreasing the Kondo peak in a continuous fashion. Therefore, by continuity with the unitary case, the Kondo effect that we detect at finite (but large) dissipation should not be an artifact of the NCA. We have added at line 278 that we only consider the occupied impurity regime $\varepsilon_d < \mu$.

2. *The response to point 3 is intriguing. I am unaware of the Langreth relation and would have guessed that such an explicit relation can only hold in equilibrium at zero temperature. But of course, I accept the point of the authors that the occupation deviates considerably from $n_\sigma \approx 1/2$. Given that the Kondo model is particle-hole symmetric, I expect G to be symmetric around that point. The Langreth relation is not symmetric around $n_\sigma = 1/2$.*

The relation that we quoted as Langreth’s relation [1] might be known to the Referee as (generalized) Friedel sum rule, and can be found in many standard textbooks (e.g., unnumbered equation after eq. 11.2.20 of Mahan’s book [2], eq. 16.76 in Coleman’s [3] and eq. 5.50 of Hewson’s [4]), as well in the already cited [5] (eq. 41). It is a nonperturbative statement that fixes the value of the impurity spectral function at the bath chemical potential (hence the linear-response conductance $G \propto A(\mu)$) at equilibrium and zero temperature. We have actually slightly misquoted the formula in the first response—the correct expression being $G = 2e^2/h \cdot \sin^2(\pi \langle n_\sigma \rangle)$. Regardless of the typo, the resulting function $G(\langle n_\sigma \rangle)$ is indeed symmetric around $\langle n_\sigma \rangle = 1/2$, since the latter value corresponds to the maximum of the sine.

3. *The response to point 4 is also interesting. Again, my naive take is that the Keldysh technique can be applied in equilibrium and then will be equivalent to the Matsubara technique. I read your reply as saying that in that case (for the NCA) there will be a difference in the diagrammatic structure of the approximation. I always thought that a rigorous projection of pseudoparticle propagators leads to objects that are very close to lesser and larger functions (instead of retarded and advanced) due to the removal of back propagation in the projection.*

While the Keldysh approach is indeed equivalent to the Matsubara one at equilibrium, in our opinion the comparison between the two is not completely straightforward, since the Matsubara Green’s function is not a combination of the retarded/advanced and lesser/greater functions that are the main objects of the Keldysh formalism. The analytic continuation that is required to go from Matsubara to retarded Green’s functions is likely the source of the apparent inconsistency. A qualitative reconciliation of the two approaches might go as follows. In the the Keldysh

approach, one separates the spectral properties, encoded in G^R , from the statistical information, encoded in $G^<$. The associated self-energies are given by diagrams that are topologically identical to the ones in the Matsubara formalism, but acquire different analytical expressions once they are “read” on the Keldysh contour. In the nonequilibrium NCA that we employed, only the lesser Green’s functions carry the “gauge charge” Q —after all, the projection involves the occupancies of the slave particles’ levels rather than their excitation energies. Hence, the constraint leads to further reducing the number of diagrams, depending on the type of self-energy that one considers (retarded or lesser). On the other hand, the Matsubara Green’s functions encode both statistical and spectral properties at the same time, hence they carry gauge charge and are subject to the same constraints on the pseudoparticle loops as the lesser Green’s functions, in accordance with the Referee’s expectation.

References

- [1] David C. Langreth. “Friedel Sum Rule for Anderson’s Model of Localized Impurity States”. In: *Phys. Rev.* 150 (2 Oct. 1966), pp. 516–518. DOI: [10.1103/PhysRev.150.516](https://doi.org/10.1103/PhysRev.150.516). URL: <https://link.aps.org/doi/10.1103/PhysRev.150.516>.
- [2] Gerald D. Mahan. *Many-particle physics*. 2nd ed. New York: Plenum Publishing Corporation, 2013.
- [3] Piers Coleman. *Introduction to Many-Body Physics*. Cambridge University Press, 2015. DOI: [10.1017/CB09781139020916](https://doi.org/10.1017/CB09781139020916).
- [4] Alexander Cyril Hewson. *The Kondo Problem to Heavy Fermions*. Cambridge Studies in Magnetism. Cambridge: Cambridge University Press, 1993. DOI: [10.1017/CB09780511470752](https://doi.org/10.1017/CB09780511470752).
- [5] Ned S. Wingreen and Yigal Meir. “Anderson model out of equilibrium: Noncrossing-approximation approach to transport through a quantum dot”. In: *Phys. Rev. B* 49 (16 Apr. 1994), pp. 11040–11052. DOI: [10.1103/PhysRevB.49.11040](https://doi.org/10.1103/PhysRevB.49.11040). URL: <https://link.aps.org/doi/10.1103/PhysRevB.49.11040>.